# A comparative analysis of soil physicochemical properties and microbial community structure among four shelterbelt species in the northeast China plain

Jia Yang,[1] Dang Ding,[1] Xiuru Zhang,[1] Huiyan Gu[1]

**ABSTRACT**  Conducting studies that focus on the alterations occurring in the soil microbiome within protection forests in the northeast plain is of utmost importance in evaluating the ecological rehabilitation of agricultural lands in the Mollisols region. Nevertheless, the presence of geographic factors contributes to substantial disparities in the microbiomes, and thus, addressing this aspect of influence becomes pivotal in ensuring the credibility of the collected data. Consequently, the objective is to compare the variations in soil physicochemical properties and microbial community structure within the understory of diverse shelterbelt species. In this study, we analyzed the understory soils of *Juglans mandshurica* (Jm), *Fraxinus mandschurica* (Fm), *Acer mono* (Am), and *Betula platyphylla* (Bp) from the same locality. We employed high-throughput sequencing technology and soil physicochemical data to investigate the impact of these different tree species on soil microbial communities, chemical properties, and enzyme activities in Mollisols areas. Significant variations in soil nutrients and enzyme activities were observed among tree species, with soil organic matter content ranging from 49.1 to 67.7 g/kg and cellulase content ranging from 5.3 to 524.0 µg/d/g. The impact of tree species on microbial diversities was found to be more pronounced in the bacterial community (Adnoism: R = 0.605) compared to the fungal community (Adnoism: *R* = 0.433). The linear discriminant analysis effect size (LEfSe) analysis revealed a total of 5 (Jm), 3 (Bp), and 6 (Am) bacterial biomarkers, as well as 2 (Jm), 6 (Fm), 4 (Bp), and 1 (Am) fungal biomarker at the genus level (LDA3). The presence of various tree species was observed to significantly alter the relative abundance of specific microbial community structures, specifically in Gammaproteobacteria, Ascomycota, and Basidiomycota. Furthermore, environmental factors, such as pH, total potassium, and available phosphorus were important factors influencing changes in bacterial communities. We propose that Fm be utilized as the primary tree species for establishing farmland protection forests in the northeastern region, owing to its superior impact on enhancing soil quality.

**IMPORTANCE**  The focal point of this study lies in the implementation of a controlled experiment conducted under field conditions. In this experiment, we deliberately selected four shelterbelts within the same field, characterized by identical planting density, and planting year. This deliberate selection effectively mitigated the potential impact of extraneous factors on the three microbiomes, thereby enhancing the reliability and validity of our findings.

**KEYWORDS**  revegetation, Mollisols, sequencing, soil management, enzyme activity, microbial diversity

Address correspondence to Huiyan Gu, ghuiyan@nefu.edu.cn.

The authors declare no conflict of interest.

See the funding table on p. 15.

Mollisols are the most fertile soil in the world, and their productivity is much higher than loess and red soil (1). The Mollisols located in the northeastern plains of China is one of the four Mollisols regions in the world, and it is also an essential commercial grain base in China. However, as human demand for nature far exceeds the protection of the environment, the problem of soil degradation is becoming increasingly severe, seriously threatening the region's agricultural production and ecological environment (2). To protect food and ecological security, China has developed a series of policy documents to provide legal solid measures for Mollisols conservation, including the Mollisols conservation and utilization tasks proposed during the 14th Five-Year Plan and the "Law of the People's Republic of China on Mollisols Conservation" published in 2022 (3). Globally, there are two mainstream approaches to soil conservation and utilization, i.e., conservation tillage and fallow crop rotation systems (4). Among them, the farmland shelterbelt system (FSS) in conservation tillage is widely used in China.

The establishment of FSS can improve the soil properties and increase the species diversity in the area, thus effectively restoring the ecological quality of the area (5). The tree species mainly determine the structural characteristics of shelterbelts but will change as the vegetation grows. Poplar is the most dominant protective forest species in northeast China due to its fast-growing features, and several studies have reported that poplar has significant improvement effects on soil and water conservation and ecosystems in agricultural fields (6, 7). However, with the development of poplar monoculture, soil problems such as sloughing and acidification occur (8); thus, studying the effect of other protection forests on soil improvement is urgent. In the context of China's Three North Protective Forest System construction project, 36 major afforestation species were released in the northeast region, including *Juglans mandshurica* (Jm), *Fraxinus mandschurica* (Fm), *Acer mono* (Am), and *Betula platyphylla* (Bp), in the current study. Broadleaf forests have been reported to have higher soil quality than coniferous and mixed coniferous and broadleaf forests, with Jm possessing higher soil fertility (9). In addition, Bp has also been shown to affect soil nutrients and enzyme activities positively (10). Interestingly, compared to Jm and Am, Fm is more favorable to soil properties due to its high-quality fine roots (11). Therefore, studying the four shelterbelt soils' characters at the same field condition is crucial for FSS in the northeast China plain.

With the development of sequencing technologies, the driving role of the microbiome in soil function has been gradually recognized, and research approaches have shifted from material cycle characterization to microbiota response (12). For example, microbes can respond to soil degradation by regulating the abundance of taxa that react to the carbon cycle (13), and mixed microbial communities strongly degrade organic compound-contaminated soil (14, 15). In addition, many nitrogen-fixing, mycorrhizal fungi promote plant growth by increasing the nutrient uptake capacity of the root system (16, 17). Vegetation type has been found to play an essential role in soil microbial community construction at small scales (18), and driven community structure has a lasting impact on subsequent farming (19). Differences in plant apoplastic nutrients and root secretions lead to different microbial communities shaped by the energy provided by plants (20). Therefore, studying the response of microbial communities to different tree species helps us to understand soil microecology. However, microbial communities are highly sensitive and intensely dynamic biological taxa influenced by geo-environmental factors such as temperature, moisture, and elevation (21). In the experimental design of this study, four tree species were planted in the same farmland at the same time, effectively eliminating the influence of geography, planting density, forest age, and other factors, which is a rarely controlled experiment under field conditions.

Accordingly, this study selected four species of shelterbelts planted for 3 years in the northeastern Mollisols area to study the composition and differences of soil microorganisms using sequencing. Meanwhile, soil chemical properties were determined to

elucidate the effects of environmental factors on microbial diversity. Finally, enzyme activities were used to characterize the amelioration effect of tree species on Mollisols. In light of the preceding discourse, we have developed three hypotheses that arise from variances in tree species: (i) the soil nutrient content exhibits variation; (ii) the composition of the soil core microbiome diverges; and (iii) the predominant flora manifest dissimilar functions, thereby resulting in disparities in their impact on soil enhancement. This study enriches the study of soil microorganisms in the Mollisols area under controlled conditions and provides technical guidance for selecting tree species for constructing farmland shelterbelts in northeast China.

## MATERIALS AND METHODS

### Site description and soil sampling

The research site, situated at Keshan Farm in the Sonnen plain (48°17′N, 125°22′E, 156 m above sea level), exhibits a topography marked by hills and scattered land, a cold-temperate subhumid monsoon climate, and the presence of typical Mollisols in the cultivated layer. The average annual precipitation and temperature recorded at this location are approximately 500 mm and 1.9℃, respectively. In April 2019, four sample plots measuring 25 m × 100 m were established on a one-hectare cropland area. Each plot was planted with identical plants spaced 3 m apart (Jm, Fm, Am, and Bp). Subsequently, in October 2022, each sample plot was randomly divided into three smaller sample squares measuring 5 m × 5 m. A five-point sampling method was employed in each sample plot to collect the top layer of bulk soil (0–10 cm) after removing surface litter and debris. A subset of the samples was then sieved and immediately frozen for DNA extraction, while the remaining soil samples were air-dried and analyzed for chemical properties and enzymatic activity. In addition, tree height and diameter at breast height (DBH) were measured as indicators of plant growth.

### Determination of soil physicochemical and enzymatic activity

Soil bulk density was determined using the ring knife method. Soil pH was measured from a 1:2.5 soil-to-water ratio using a pH meter. Total carbon (TC) was determined by the TOC analyzer (Model multi N/C 2100S, Analytik Jena). Soil organic matter (SOM) was measured using the potassium dichromate-external heating method. Total nitrogen (TN) was determined using the Kjeldahl method, and available nitrogen (AN) was determined using the alkaline dissolved diffusion method. Total phosphorus (TP) was measured using the concentrated sulfuric acid and perchloric acid digestion-molybdenum antimony colorimetry method. Available phosphorus (AP) was measured using hydrochloric acid and the ammonium fluoride extraction-molybdenum antimony colorimetry method. Total potassium (TK) was determined using the concentrated sulfuric acid, perchloric acid digestion-flame photometric method. Available potassium (AK) by ammonium acetate leaching-flame photometric method. Additionally, five soil enzymes associated with organic carbon, nitrogen, and phosphorus were quantified using the Soil-Urease kit, Soil Acid Phosphatase kit, Soil Dehydrogenase kit, Soli-Cellulase (S-CL) kit, and Soil-Sucrase kit. The kits were procured from Nanjing Jiancheng Bioengineering Institute in China. Enzyme activity was assessed at wavelengths of 578, 405, 540, 540, and 485 nm using a microplate reader (SpectraMax iD3, Molecular Devices).

### DNA extraction, high-throughput sequencing, and data processing

Genomic DNA extraction was performed utilizing a commercially available kit (ALFA-SEQ Advanced Soil DNA) following the manufacturer's instructions. DNA concentration and purity were determined using Qubit 3.0 and Nanodrop One instruments (Thermo Fisher Scientific, Waltham, USA). The V3–V4 region of the bacterial 16S rRNA gene and the

ITS1 region of the fungal ITS gene were amplified using the TaKaRa Premix Taq Version 2.0 (TaKaRa Biotechnology Co., Dalian, China) kit with genomic DNA as a template and specific primers 338F-806R and ITS1F-2043R, respectively. The length and concentration of the PCR product were determined through 1% agarose gel electrophoresis. Samples displaying a prominent band within the targeted regions were selected for further experimentation. The PCR products were combined in equal density ratios based on the GeneTools Analysis Software (Version 4.0, SynGene). Next, the PCR product mixture was purified using the EZNA Gel Extraction Kit (Omega, USA). Library construction was conducted by generating sequencing libraries with the NEBNext Ultra II DNA Library Prep Kit for Illumina (New England Biolabs, MA, USA), following the manufacturer's guidelines and incorporating index codes. The library's quality was evaluated using the Qubit@ 2.0 Fluorometer (Thermo Fisher Scientific, MA, USA). Finally, the library was sequenced on an Illumina Nova6000 platform, generating 250 bp paired-end reads (Guangdong Magigene Biotechnology Co., Ltd. Guangzhou, China).

The quality control of the Raw Reads was performed using Fastp (version 0.14.1) with a sliding window (-W 4 -M 20). The removal of primers was carried out using Cutadapt software, taking into account the primer information located at the beginning and end of the sequence. This process resulted in obtaining the paired-end Clean Reads. Considering the overlapping relationship between the paired-end reads, the paired-end clean reads were merged using usearch—fastq_mergepairs (V10). Specifically, when there was an overlap of at least 16 bp, the read generated from the opposite end of the same DNA fragment was merged. The maximum allowed mismatch in the overlap region was 5 bp, and the spliced sequences were called Raw Tags. Raw Tags are again passed through Fastp's quality control to obtain the paired-end Clean Tags (22). The whole genome sequencing data generated in this study have been submitted to the NCBI SRA database (https://www.ncbi.nlm.nih.gov/bioproject/) under accession number PRJNA1047323.

We employed UPARSE, a clustering method with a 97% identity threshold, and DADA2, a denoising method, to process the 16S rRNA and ITS gene amplicon fragment data sets (23, 24). This allowed us to generate operational taxonomic units (OTUs) and amplified sequence variants (ASVs) for each data set. Taxonomic information for each representative sequence was annotated using the Silva database (for 16S rRNA) and the Unite database (for ITS) through the use of the usearch—sintax tool, with a confidence threshold set to default (≥0.8). The species annotation taxonomy was categorized into seven levels: kingdom (L1), phylum (L2), class (L3), order (L4), family (L5), genus (L6), and species (L7). During the clustering process, usearch was employed to eliminate both the chimera sequence and singleton OTUs simultaneously. Furthermore, OTUs annotated as chloroplasts or mitochondria (16S rRNA amplicons) and those that could not be assigned to the kingdom level were excluded. Subsequently, the OTU and ASV taxonomy synthesis information table was obtained for subsequent analysis.

## Statistical analysis

The microbial diversity assessment was conducted by utilizing the Simpson index and nonmetric multidimensional scaling (NMDS) in conjunction with the ANOSIM test (bray_curtis) (25). The relative abundance of phylum and genus levels was employed to depict the composition of the microbial community. To examine differences in biomarker abundance between multiple groups, linear discriminant analysis effect size (LEfSe) was employed, with a significance threshold set at LDA > 3, using the Kruskal–Wallis rank sum test (26). Additionally, redundancy analysis (RDA) was conducted to elucidate the impact of soil physicochemical properties on the microbial community (27). The significance of this analysis was determined through Monte Carlo permutation (permu = 999). This approach further explored the interrelationship between environmental factors and microbial flora. The Spearman correlation coefficients between environmental factors and species abundance were computed and visually represented

through a heat map (27). All statistical analyses were performed utilizing SPSS version 22.0, and the significance of differences was analyzed by one-way ANOVA followed LSD post hoc mulitiple comparison tests ($P < 0.05$).

## RESULTS

### Shelterbelt and soil biogeochemical parameters

The four shelterbelts in this study were planted on the same cultivated area (48°17′N, 125°22′E), effectively controlling for differences in soil properties brought about by geographical location and environmental factors, which prompted us to focus more attention on the effects of stand differences on the soil (Fig. 1). The results for DBH and tree height showed that Jm had higher stand volume. In contrast, the Am possessed the lowest stand volume under the same conditions of stand age, and there were significant differences among the four stands ($P < 0.05$; Table 1).

The physicochemical properties of the samples collected from different shelterbelts differed significantly ($P < 0.05$; Table 1). Regarding physical structure, there were no significant differences between the three groups, except for the slightly lower bulk weight of Bp (1.27 g/cm$^3$). Another important indicator of soil pH is that all measurements were somewhat acidic (6.07–6.39). Regarding the soil chemical properties, the results showed that the content of Fm was significantly higher than the other groups, except for the data related to K content. The Jm group had the best above-ground growth and had the highest K content in the understory soil (TK = 1.97 g/kg, AK = 26.49 mg/kg). In addition, Bp, which had the weakest above-ground growth, had the lowest soil N, P, and K contents, except for a significantly lower C content (TC = 44.29 g/kg, SOM = 52 g/kg) than the other groups.

This study further analyzed the soil's biological activity and nutrient transformation by measuring the five enzyme activities. Fm was significantly higher in the five enzymes than the other groups, and the most prominent cellulase content (523.99 µg/d/g) was about 10 times higher than Bp (48.28 µg/d/g) and 100 times higher than Jm (5.28 µg/d/g) (Fig. 2D). Bp had the most elevated acid phosphatase and dehydrogenase among the four shelterbelts with 779.25 nmol/h/g and 38.89 µg/d/g, respectively, but the lowest sucrase (100.52 mg/d/g). Jm had the lowest urease and dehydrogenase with 1174.80 and 11.67 µg/d/g, respectively. Soil enzyme activity of Am, similar to above-ground growth, was in the middle of the four shelterbelts, possessing only the lowest acid phosphatase (589.27 nmol/h/g) (Fig. 2B). To summarize, it can be observed

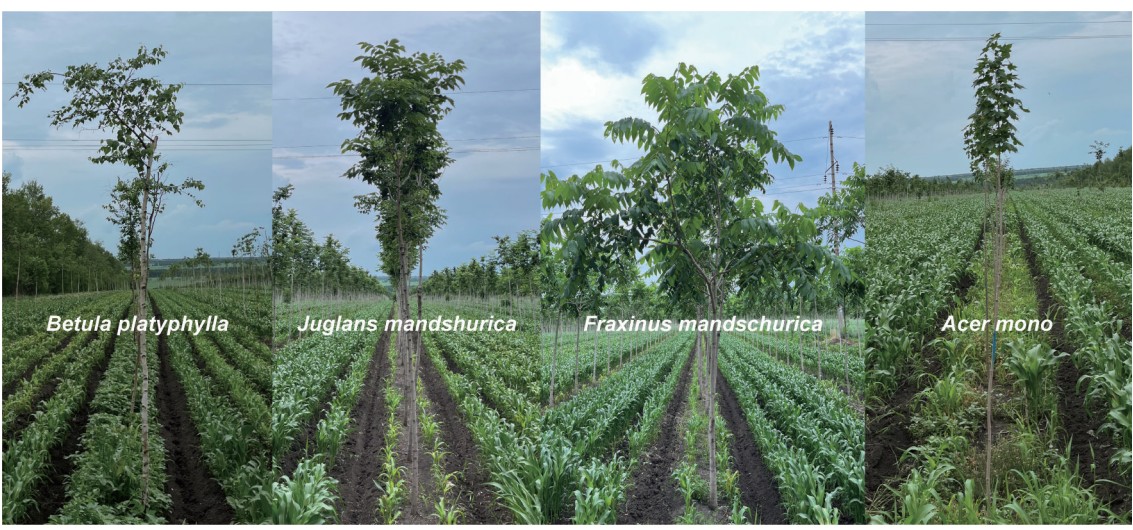

*Betula platyphylla*   *Juglans mandshurica*   *Fraxinus mandschurica*   *Acer mono*

**FIG 1** Shelterbelts planted and tree growth situation in this study.

**TABLE 1** Summary of the environment parameters analyzed by LSD post hoc comparison tests[a]

| | Jm | Fm | Am | Bp |
|---|---|---|---|---|
| Plant parameters | | | | |
| Tree height (m) | 3.59 ± 0.06ab | 3.77 ± 0.14a | 2.14 ± 0.06c | 3.34 ± 0.12b |
| Diameter at breast height (mm) | 45.24 ± 1.45a | 33.75 ± 1.11b | 8.07 ± 0.54d | 27.73 ± 0.61c |
| Soil biogeochemical parameters | | | | |
| Bulk density (g/cm³) | 1.29 ± 0.01ab | 1.31 ± 0.01a | 1.29 ± 0ab | 1.27 ± 0.02b |
| pH | 6.07 ± 0.03c | 6.28 ± 0.03ab | 6.11 ± 0.1bc | 6.39 ± 0.04a |
| Total carbon (g/kg) | 48.59 ± 2.26bc | 57.07 ± 1.82a | 44.29 ± 1.15c | 52.14 ± 1.82ab |
| Organic matter (g/kg) | 49.1 ± 1.75c | 67.73 ± 2.01a | 52 ± 1.25c | 58.68 ± 0.92b |
| Total nitrogen (g/kg) | 1.35 ± 0.06b | 1.63 ± 0.05a | 1.41 ± 0.03b | 1.46 ± 0.01b |
| Alkali nitrogen (mg/kg) | 231.47 ± 7.5b | 252.68 ± 4.51a | 238.82 ± 3.2ab | 240.42 ± 2.89ab |
| Total phosphorus (g/kg) | 0.58 ± 0.02c | 0.9 ± 0.03a | 0.7 ± 0.02b | 0.69 ± 0.05b |
| Available phosphorus (mg/kg) | 29.8 ± 3.57b | 42.34 ± 3.44a | 35.51 ± 3.42ab | 36.02 ± 3.76ab |
| Total potassium (g/kg) | 1.97 ± 0.15a | 1.5 ± 0.06b | 1.62 ± 0.1b | 0.88 ± 0.03c |
| Available potassium (mg/kg) | 26.49 ± 0.38a | 11.83 ± 1.82b | 24.39 ± 1.37a | 8.41 ± 1.54b |

[a]Different letters represent significant differences (abcd, $P < 0.05$).

that Fm exhibited the most prominent soil enzyme activity and chemical nutrient levels compared to the other shelterbelt species.

## Comprehensive characterization of microbial community composition

To explore the comprehensive characterization of the soil microbiota, this study included 24 microbial diversities sampled from four soil habitats (Jm, Fm, Am, and Bp) (Table S1). After quality control, 16S rRNA and ITS gene sequencing obtained 1,385,941 and 945,934 clean tags, respectively. A total of 9,986 bacterial OTUs and 2,352 fungal ASVs were annotated. Venn diagrams calculated based on OTU abundance showed that 31.96% (3,192/9,986) of bacterial OTUs and 9.52% (224/2,352) of fungal ASVs were common in four shelterbelt soils (Fig. 3A and B). Among the bacterial communities within the soil samples, it was observed that Jm exhibited the highest number of unique OTUs (1,401), whereas Bp displayed the lowest number (751). Regarding fungi, Fm-specific ASVs were found to be the most abundant (488), while Am's ASVs were the least abundant (354). The α-diversity was related to the diversity and richness of the microbial community. The results showed that the Simpson index of Bp was significantly higher in bacteria than in the other three species (Fig. 3C). At the same time, in fungi, there was no significant difference between the four species (Fig. 3D). NMDS analysis reflecting microbial community structure showed that the microbial composition was significantly

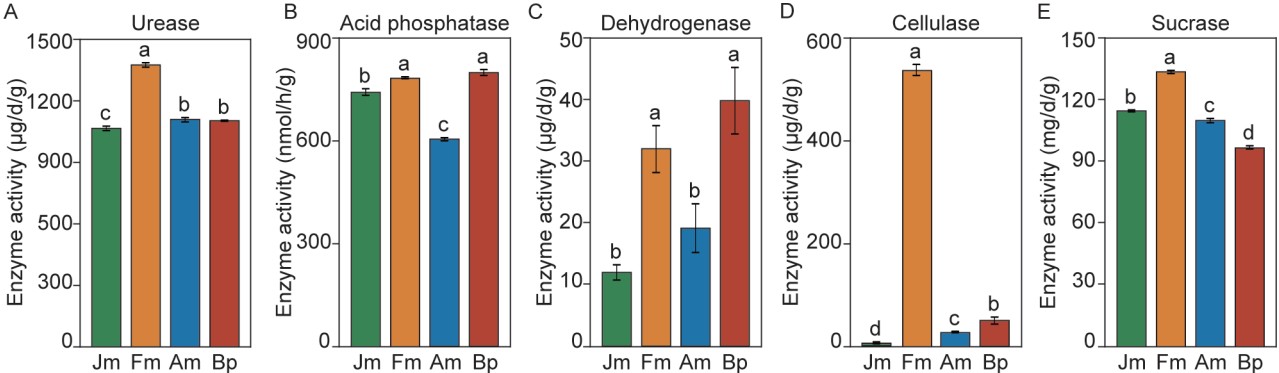

**FIG 2** The effects of tree species on the activity of urease (A), acid phosphatase (B), dehydrogenase (C), cellulase (D), and sucrase (E). LSD post hoc comparison tests analyzed the significant differences; different letters represent significant differences (abcd, $P < 0.05$). The error bars indicate the standard deviation of three replicates.

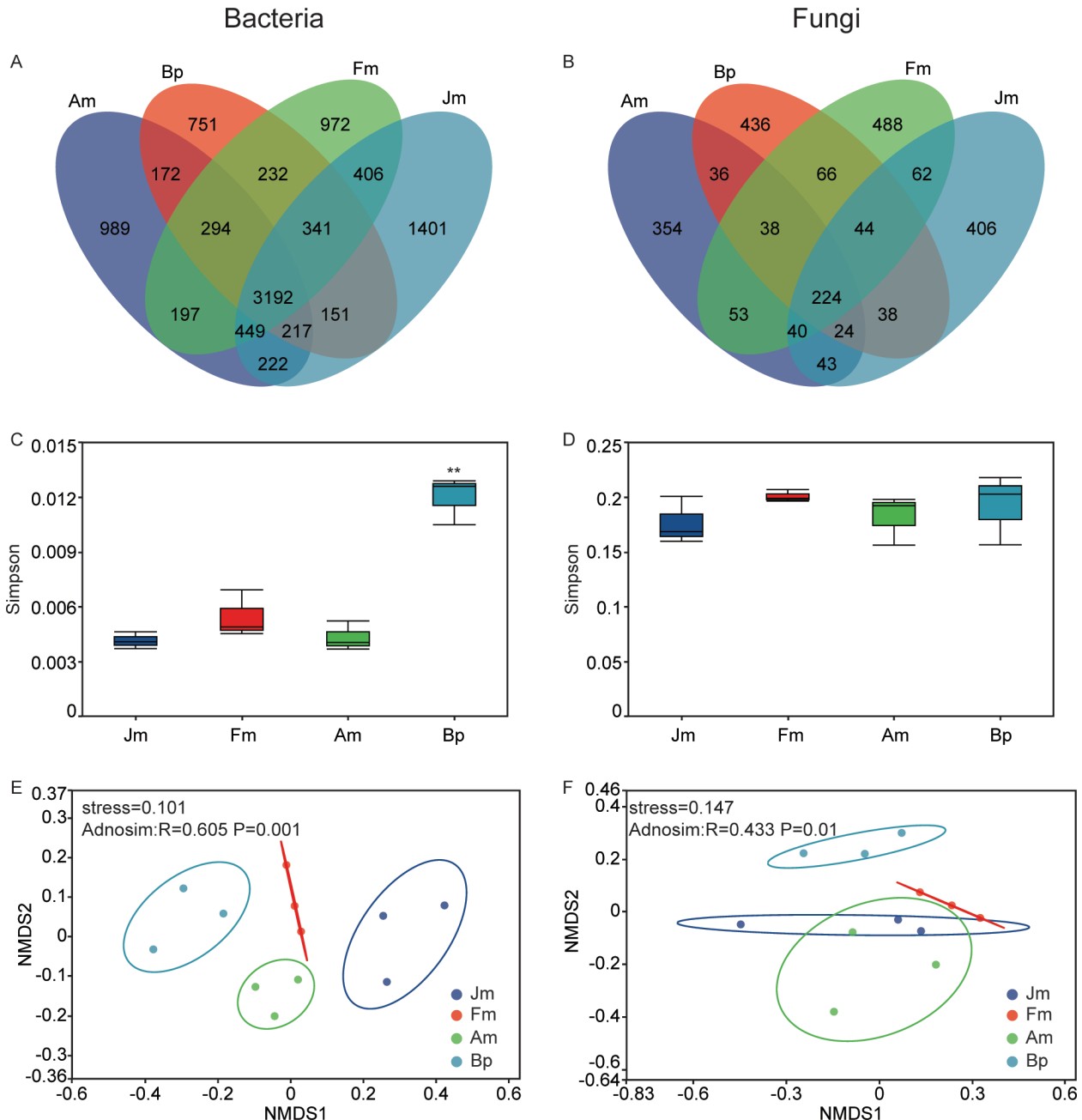

**FIG 3** Venn, alpha diversity, and beta diversity analysis of microbes on different tree species. Venn analysis of the bacteria's OTUs (A) and fungi's ASVs (B). Simpson indices of the bacterial (C) and fungal (D). NMDS plots of the bacterial (E) and fungal (F). The standard deviation was shown at the top of the bar graph, and the asterisk in the figure represented a significant difference calculated by the Student's $t$-test (**$P < 0.01$).

different (Fig. 3E and F) among soils in four shelterbelts ($P < 0.01$), with a more significant difference within the bacterial group ($R = 0.605$) than in the fungal ($R = 0.433$).

Additionally, the relative abundance of microbial composition was analyzed at both the phylum and genus levels. The dominant phyla among the bacterial communities in all soil samples were identified as Acidobacteria (25.68%), Proteobacteria (21.67%), Actinobacteria (13.50%), Verrucomicrobia (10.86%), Bacteroidetes (8.64%), Chloroflexi (6.02%), and Gemmatimonadetes (5.60%). These phyla collectively accounted for approximately 92% of all bacterial species (Fig. 4A; Table S2). Other taxa with an abundance greater than 1% included Rokubacteria (2.30%), Planctomycetes (1.10%), and

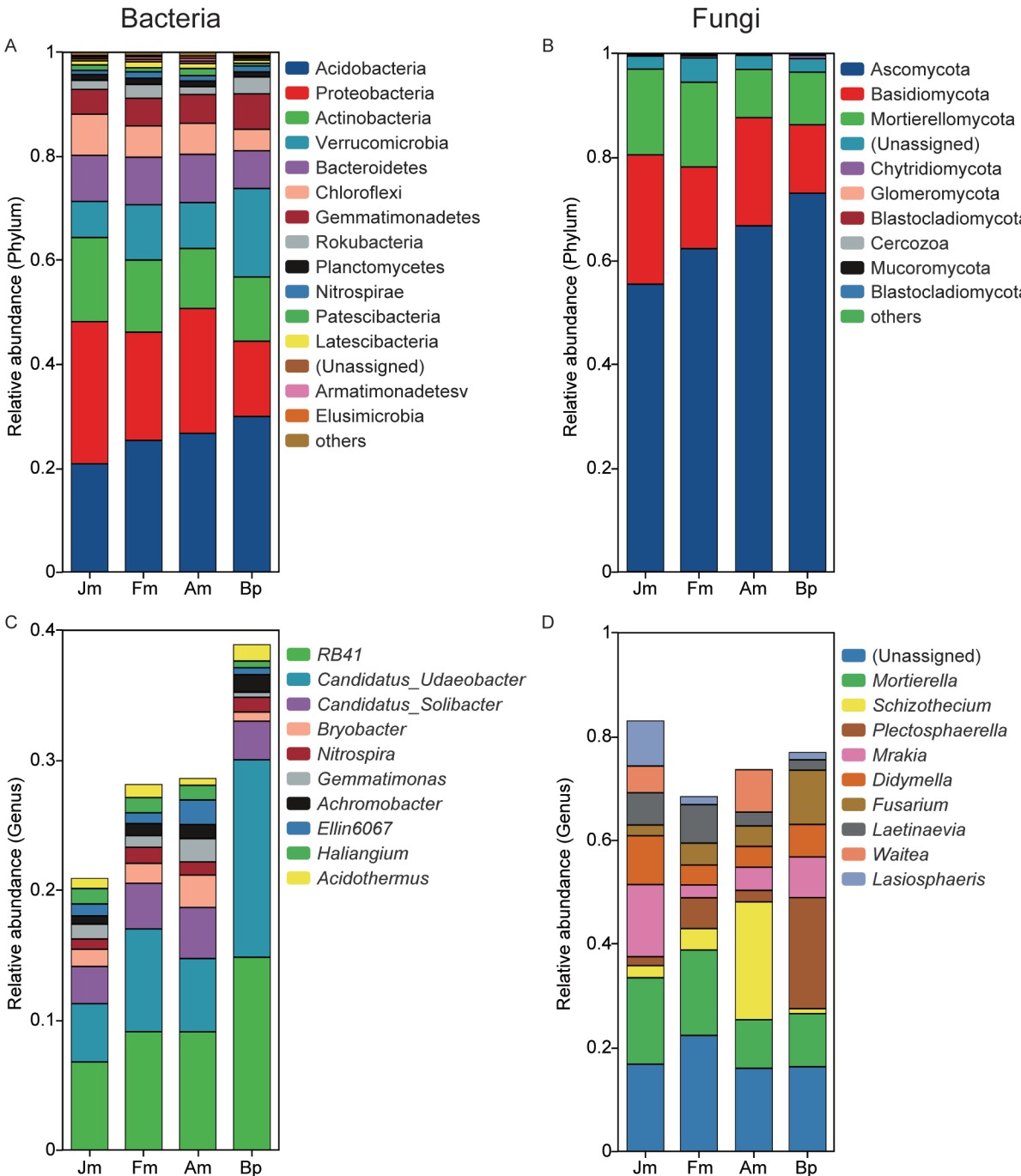

**FIG 4** Microbial community composition of microbes on different tree species. The relative abundances at the phylum level of the bacterial (A) and fungal (B).The relative abundances at the genus level of the bacterial (C) and fungal (D).

Nitrospirae (1.04%). Acidobacteria (29.92%), Verrucomicrobia (17.04%), Gemmatimona-detes (6.80%), and Rokubacteria (3.27%) were found to be significantly more abundant in Bp compared to the other groups, while Jm's Proteobacteria (27.35%), Actinobac-teria (16.21%), and Chloroflexi (7.92%) exhibited significantly higher abundance than the other groups. At the genus level, *RB41* (9.95%), *Candidatus_Udaeobacter* (8.31%), *Candidatus_Solibacter* (3.32%), *Bryobacter* (1.52%), *Nitrospira* (1.04%), *Gemmatimonas* (1.06%), *Ellin6067* (1.05%), *Achromobacter* (1.00%), *Haliangium* (1.00%), and *Acidothermus* (0.92%) were the top 10 bacterial taxa, collectively representing approximately 29% of all

sequences (Fig. 4C). Furthermore, notable variations in the relative abundance of the top 10 genera were observed among the four shelterbelts soils (Table S3).

In the fungal community, most sequences at the phylum level were identified as Ascomycota (64.43%). Additionally, the top three most abundant taxa included Basidiomycota (18.72%) and Mortierellomycota (13.09%), collectively representing approximately 96.24% of all fungal species (Fig. 4B; Table S2). Interestingly, the abundance of Ascomycota was significantly higher in Am (66.77%) and Bp (73.07%) compared to Jm (55.50%) and Fm (62.37%), while the abundance of Mortierellomycota was significantly lower in Am (9.31%) and Bp (10.16%) compared to Jm (16.55%) and Fm (16.35%). At the genus level, the top 10 fungal taxa were *Unassigned* (17.97%), *Mortierella* (13.09%), *Schizothecium* (7.50%), *Plectosphaerella* (7.76%), *Mrakia* (7.12%), *Didymella* (5.84%), *Fusarium* (5.15%), *Laetinaevia* (4.53%), *Waitea* (3.32%), and *Lasiosphaeris* (2.94%) (Fig. 4D), accounting for approximately 75.14% of all sequences. Notably, *Schizothecium* exhibited the most significant variation in abundance among the four samples, ranging from 0.96% to 22.58% (Table S3).

A total of 46 bacterial biomarkers and 24 fungal biomarkers (LDA3, > $P$ < 0.05) were screened by LEfSe analysis from phylum to genus level (Fig. 5). Among them, no bacterial biomarker was identified in Fm. The findings of the study indicate that Proteobacteria was the sole bacterial phylum observed in Jm, with *Pseudomonas*, *Pseudarthrobacter*, *Acidibacter*, *Niastella*, and *Leptolyngbya_EcFYyyy* identified as biomarkers at the genus level. Similarly, Verrucomicrobia was identified as the only biomarker bacterial phylum in Bp, while *Candidatus_Udaeobacter* and *Waddlia* were identified as biomarkers at the genus level. In contrast, no phylum-level biomarker was identified in Am, but six biomarkers were recognized at the genus level, including *Bryobacter*, *Ellin6067*, *Gemmatimonas*, *Massilia*, *Sphingomonas*, and *MND1* (Fig. 5A; Table S4). In addition, no biomarker at the fungal phylum level was identified in any of the four samples. However, at the genus level, *Volutella* and *Entoloma* were identified as biomarkers for Jm, *Metarhizium*, *Vishniacozyma*, *Tylospora*, *Phialocephala*, *Clitopilus*, and *Amanita* were identified as biomarkers for Fm, and *Plectosphaerella*, *Ramophialophora*, *Leucoagaricus*, and *Mallocybe* were identified as biomarkers for Bp. Lastly, Am had only one biomarker identified as *Schizothecium* (Fig. 5B; Table S5). In a word, Jm has the most numerous and diverse biomarkers.

## Correlation analysis of microbial and environmental factors

After characterizing environmental factors and microbial composition structure, we used RDA and correlation methods for data analysis to further find the intrinsic link between the two. RDA results showed that environmental factors were significantly positively correlated with differences in community structure, with bacterial structure variation on both ordination axes modeled at 80.9% and 10.5% of the explanatory degree (Fig. 6A). Bacterial communities clustered between Fm and Bp on the first RDA axis, while Fm and Am clustered on the second RDA axis. pH, TK, and AK had a more significant effect on bacterial community differences than other environmental factors. The environmental factors explained 64.7% of the cumulative structural differences of fungal communities on both axes, which was significantly lower than that of bacteria, and the distribution of the four groups of samples on the graph was also more dispersed than that of bacteria. In addition to TK and AK, TC was the main environmental factor that influenced the differentiation of fungal communities (Fig. 6B).

The correlation results showed that environmental factors were more strongly associated with bacterial genera than fungi. Specifically, K showed highly significant positive correlations with *Bradyrhizobium*, *Gemmatimonas*, and *Sphingomonas* ($P$ < 0.01) and highly significant negative correlations with *Candidatus_Udaeobacter* and *RB41* ($P$ < 0.01), while pH showed an opposite pattern of correlation with these five bacterial genera than K. In addition, the bacteria that showed a significant positive correlation with environmental factors was *Acidothermus*, and a significant negative correlation was *Massilia* (Fig. 6C). Interestingly, a limited number of fungi exhibited substantial

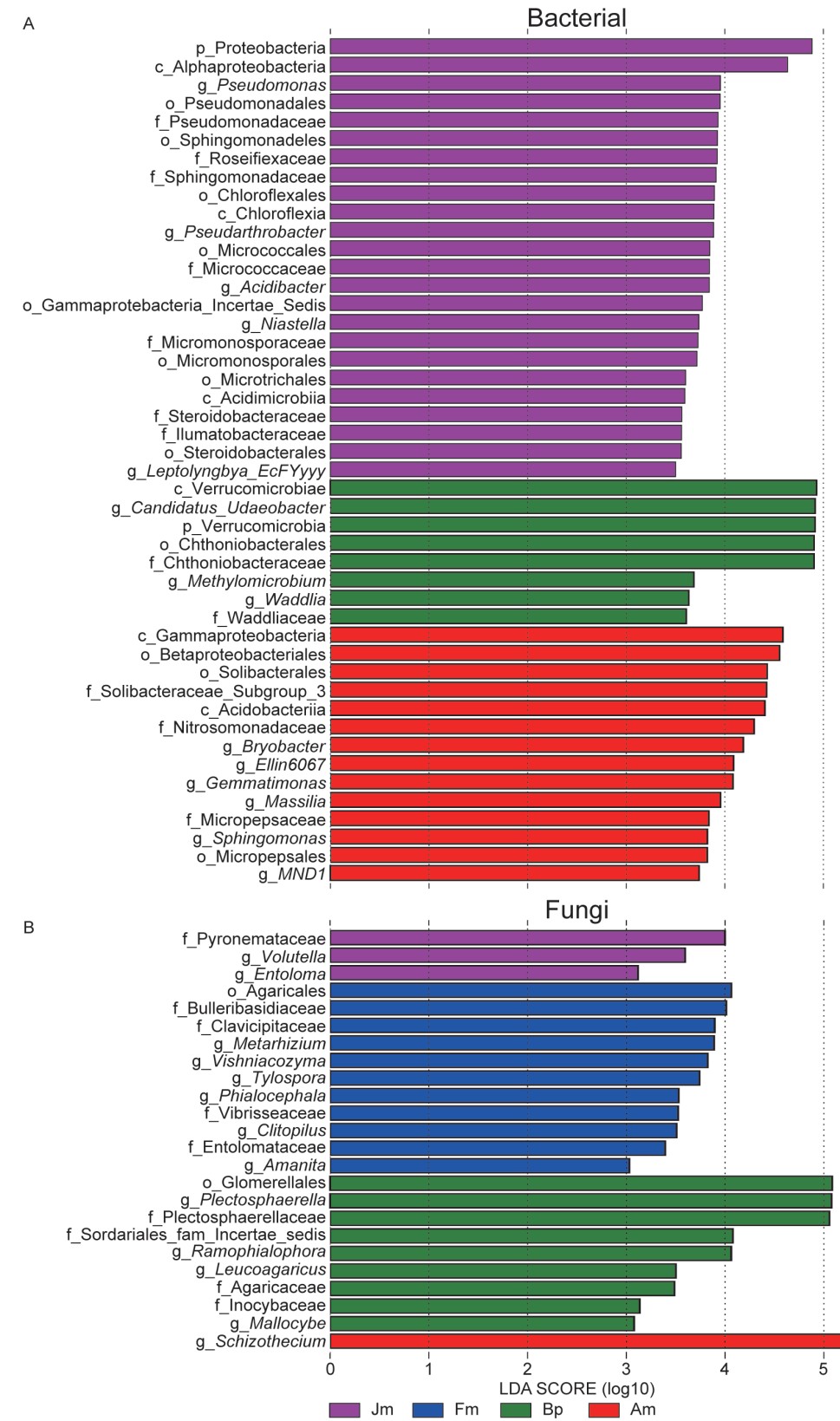

**FIG 5** LEfSe analysis of microbes on different tree species. The phylum level to genus level of the bacterial (A) and fungal (B).

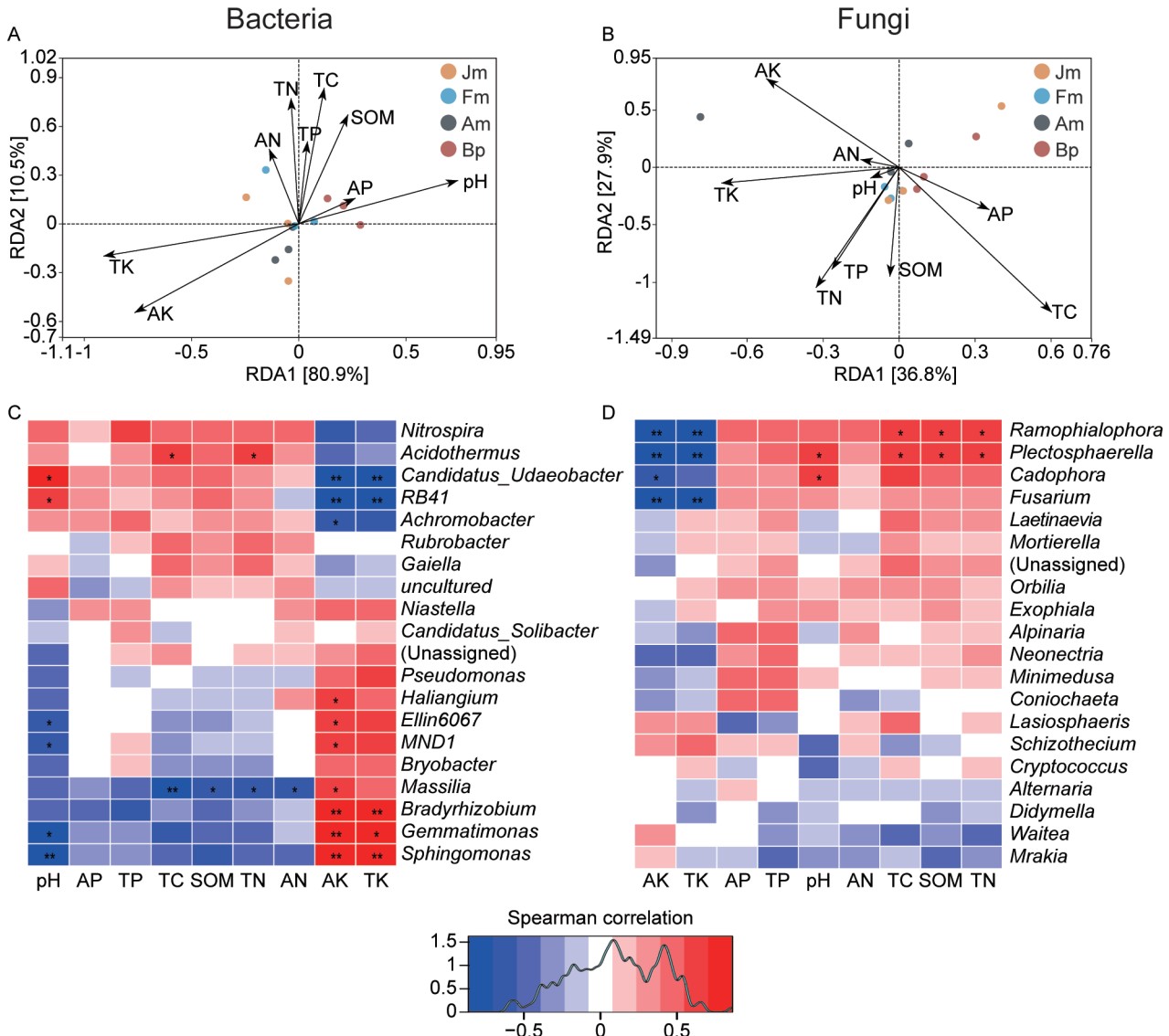

**FIG 6** RDA demonstrates shifts in different tree species affected by soil properties at the genus level of the bacterial (A) and fungal (B). Heatmap of Spearman's rank correlation coefficients combined with cluster analysis of the top 20 bacterial (C) and fungal (D) genus-level relative abundance (%, *n* = 3) between different species. *Note*: The horizontal row represents soil chemical properties, the vertical row represents microbial community abundance information, blue represents negative correlation, red represents positive correlation, the darker color indicates a higher correlation, and the *P*-value is the correlation test result (**P* < 0.05, ***P* < 0.01). AP, available phosphorus; TP, total phosphorus; TC, total carbon; SOM, soil organic matter; TN, total nitrogen; AN, available nitrogen; AK, available potassium; TK, total potassium.

correlations with environmental factors. Likewise, element K displayed highly significant negative correlations within the fungal community with Ramophialophora, Plectosphaerella, Cadophora, and Fusarium (*P* < 0.05). Conversely, *Plectosphaerella*, *Cadophora*, and *Fusarium* demonstrated significant positive correlations with pH, TC, SOM, and TN (*P* < 0.05) (Fig. 6D). Ultimately, it can be concluded that the alteration of soil chemistry properties had a more significant impact on the bacterial community compared to the fungi.

## DISCUSSION

The growth process of shelterbelts affects the soil in many ways, including physical, chemical, and biological. With the advancement of sequencing technology, more

and more reports show that environmental microorganisms are central to soil function. Therefore, this study focused on elucidating the composition and differences of understory microorganisms under different tree species under controlled conditions. In addition, soil nutrients and enzyme activities were determined in this study, and these changes were linked to microbial by correlation analysis to find the potential relationship of Mollisols recovery by different shelterbelts mediating biomarkers.

## Response of soil properties to tree species

Soil is an essential foundation of ecosystems, and above-ground vegetation types are closely related to below-ground soil fertility (28). Soil capacity is a crucial indicator of soil compactness, and its porosity is influenced by microbial activity and root growth (29). Bp exhibits a whisker root system and a rapid growth rate, with its abundant root system contributing to enhanced soil porosity (30). Consequently, Bp's soil capacity is marginally lower than other forest stands. However, given the relatively juvenile state of the shelterbelts and the constrained root development in the subterranean component of the plantation, the variation in soil capacity among the other three tree species did not exhibit statistical significance. It is postulated that this disparity may become more pronounced as the plantation matures and will persist as an observable phenomenon in forthcoming investigations. Plants alter soil properties mainly through root secretions and tree litter (31), while soil nutrients are the basis for maintaining the productivity of natural ecosystems and can directly affect plant growth and development (32). The four tree species selected in this study, all of which are common deciduous trees in the northeast, mitigate the effects of tree litter on soil properties (33). Different tree species significantly affected soil C/N/P content, with the most pronounced changes in P (Table 1). Studies have shown that phosphorus accelerates cell division and promotes faster growth of the root system and above-ground parts (34), of which Jm has the best growth among the four tree species. We suggested that the P content of Jm was significantly lower than that of the other plots, probably due to the dominance of the above-ground part in the nutrient cycling process, which resulted in more P flowing from the underground to the above-ground.

Enzyme activity is a sensitive indicator of the metabolic activity of soil microorganisms, which is not only a significant participant in the cycling (carbon, nitrogen, phosphorus, and sulfur) of soil and vegetative matter but also promotes growth by altering nutrient status and increasing the effectiveness of nutrient uptake by plants (35). Like the soil nutrient profile, soil enzyme activities were significantly higher in the Fm than in other species. This finding further confirms that soil enzyme activities are closely related to factors such as the form content of soil nutrients and an essential indicator for evaluating soil quality (36, 37). For arable land, the core of soil degradation is reducing organic matter, and glucose catalyzed by S-CL is the primary carbon source in the C cycle (38). Interestingly, S-CL activity in Fm was hundreds of times higher than in the other sample plots (Fig. 1D), and such highly significant differences are intriguing. We proposed that Fm plays a vital role in repairing the Mollisols carbon cycle, possibly due to its unique root secretion and microbiome, whereas the other four enzyme activities fluctuated by S-CL (39). In summary, this study found significant differences in the effects of different tree species on soil physicochemical properties and enzyme activities in the understory, a finding that supports hypothesis 1.

## Response of soil microbial community to tree species

The biodiversity of soil microorganisms is essential for maintaining soil quality, productivity, and ecological balance of agroecosystems (12). Based on the microbial diversity research method, it was found that the bacterial richness and diversity of Bp were significantly higher than other species, and the soil microbial structures of the four tree species were significantly different (Fig. 3). This result supports hypothesis 2 that the richness and diversity of microbial communities in the understory of varying tree species also differ (40). Soil fungi have been reported to have a more stable topological network

than bacteria (41), which explains the lesser variation in fungal community structure by tree species (Fig. 3E and F).

Acidobacteria has been reported to be the most abundant taxa in Bp (42), and the results of this study are consistent with them. However, the content of Acidobacteria in Bp was significantly higher than the other samples, which may be due to the unique root secretion of Bp that alters the root environment, which in turn recruits Acidobacteria selectively (Fig. 4A) (43). In addition, *RB41* and *Candidatus_Udaeobacter*, which are abundantly contained in Bp taxa, have been shown to have a positive role in carbon cycling and soil remediation (Fig. 4C) (44, 45). Interestingly, between the four species, Jm contained the most Proteobacteria and Chloroflexi and the least Acidobacteria, Verrucomicrobia, and Gemmatimonadetes, while the opposite was true for Bp (Table S2). In addition to bacteria, fungi are essential in soil geochemical cycles (46). As in most studies, Ascomycota, Mortierellomycota, and Basidiomycota were the soil's most dominant fungi phyla (47). The structure of the fungal community varied under different tree species (Fig. 4B), and this difference was more evident at the genus level (Fig. 4D). The vast majority of dominant fungal genera belonged to Ascomycota and Basidiomycota and were mostly saprophytic, with mutualistic solid relationships with plants (48, 49). In addition, *Mortierella*, the only one of the dominant genera under Mortierellomycota, some species promotes plant growth in agricultural soils (50).

Screening of biomarkers of the four plots by using LEfSe analysis showed that bacteria were more sensitive to changes in tree species than fungi (Fig. 5). It is noteworthy that Fm, which has the best nutrient status among the four tree species, possesses the least number of biomarkers, even has no in bacteria. This result was consistent with our previous study in the Mollisols region, where high-quality soils possessed fewer but beneficial microbiomes (13). Key taxa play unique and critical roles in microbial communities and are central drivers of soil metabolic activities and nutrient cycling (51, 52). The abundance of microbial taxa in the Jm also hints at the complexity of soil functions. For example, *Pseudomonas* as pathogens can also promote plant growth (17); Sphingomonadaceae can remediate soils by degrading Bisphenol A and thus repairing them (53); Chloroflexia and *Leptolyngbya_ EcFYyyy*'s photosynthetic autotrophic capacity contributes to material cycling in the soil microenvironment (54). In addition, actinomycetes enriched in Jm can promote plant growth by increasing nutrient utilization and improving the efficiency of the material cycle in the ecosystem (55). *Candidatus_Udaeobacter* and *Methylomicrobium*, Bp's biomarkers, predominantly function in the carbon cycling pathway (44, 56). Am's biomarkers belonged to α/β/γ-proteobacterial, which are Gram-negative bacteria that not only participate in the process of nitrification/denitrification in the soil but also fix molecular nitrogen in the air as a source of nitrogen for their growth, which is essential for agriculture (57, 58).

Compared to fungi, bacterial communities are more widely studied and applied in soil remediation. However, fungi also significantly function in soil proventing and plant nutrient cycling. *Schizothecium*, a coprophilous fungus, is the sole biomarker in Am and can be found in diverse fecal matter types. It is widely recognized as a biocontrol agent against soil-borne pathogens (59). The intricate and varied nature of microbial networks often makes it challenging to categorize the role of soil microbial communities as solely beneficial or harmful. For instance, *Volutella*, a biomarker for Jm, exhibits distinct functions in different environments. In the context of soil, *Volutella buxi* demonstrates a significant capacity for pathogenicity, causing extensive destruction (60).

Conversely, *Volutella ciliata* is a saprophyte and decomposer, contributing to the accumulation of organic matter in the soil (61). Furthermore, a separate investigation has documented the heightened predatory nematode activity of *Volutella citrinella* (62). A similar scenario unfolds in the case of Bp and Fm. *Fusarium* and *Plectosphaerella* aggregate in Bp and encompass numerous plant pathogens, including *Fusarium verticillioides* and *Plectosphaerella citrulli*. These pathogens inflict substantial harm on a wide array of crops through their pathogenicity and virulence factors (63, 64). Moreover, *Leucoagaricus*, certain fungal genera have been recognized for their ecological

functional benefits. These genera not only facilitate lignin degradation through cellulase production but also contribute to the detoxification of secondary metabolites in plant tissues via oxidoreductase production (65). Consequently, they play a crucial role in the cycling of nutrients within ecosystems and the turnover of carbon. In this study, it was also observed that under identical screening conditions, the biomarkers of Fm only consisted of fungi, with a significant proportion being probiotics that possess well-defined functionalities. For instance, the fungi *Clitopilus* and *Amanita* were found to enhance nitrogen uptake in plants through symbiotic associations with etomycorrhizal mycorrhizal (66). Additionally, *Tylospora* was observed to facilitate atmospheric nitrogen cycling by generating $N_2O$, while the yeast *Vishniacozyma* exhibited a positive impact on land improvement (67, 68). Furthermore, *Metarhizium*, enriched in Fm, is recognized as a biocontrol fungus due to its potent efficacy against pests and environmentally friendly nature. In conclusion, our findings indicate that the biomarkers present in Fm primarily consist of fungi that exhibit symbiotic relationships with plants and soil, thereby significantly contributing to nutrient cycling and soil biocontrol mechanisms.

## Response pattern of soil microbiome to tree species

Environmental factors' importance for microbial communities cannot be ignored. It has been shown that environmental factors such as pH, AK, and AP determine the composition and structure of soil microbial communities in different ecosystems (47, 69). A nutrient-rich substrate favors microbial growth, while microbial diversity is suppressed in adverse environments such as, for example, drought and salinity (41). In this study, the analysis of soil physicochemical properties showed significant differences in soil properties between different understories, and the content of soil P and K changed significantly among the four plots, affecting the uptake of nutrients required for plant and microbial growth. In addition, the conditional effect of environmental factors on the structure of bacterial communities was significantly more substantial than that of fungi by RDA and heatmap analysis (Fig. 6). Researchers found that soil fertility is the most critical soil property that regulates microbial abundance, e.g., rich organic matter can provide more nutrients, which is favorable for bacterial enrichment (49). The positive relationships with *Bradyrhizobium*, *Gemmatimonas*, and *Sphingomonas* in C/N/P cycling were also in this study further confirmed (Fig. 6C) (70–72). Furthermore, alterations in the composition of fungal communities are influenced by environmental factors. Most fungal genera that hold significance in nutrient relevance in this investigation serve as biomarkers for Bp and have been demonstrated to be pathogenic to plants (63, 64). We propose that substantial increases in the concentrations of *Fusarium*, *Cadophora*, and *Plectosphaerella* contribute to the vulnerability of plants to disease as they compete with potassium within the soil. Nevertheless, microbial functions tend to be intricate and varied. While no study has established a direct association between these functions and soil fertility, they exhibited a notable and positive correlation with TC, SOM, and TN content (Fig. 4D). This suggests that a reciprocal relationship between the enrichment of *Ramophialophora* and *Plectosphaerella* and the levels of soil carbon and nitrogen.

Based on the preceding discourse regarding the roles played by soil core microorganisms in diverse ecosystems, we present a synthesis of how microorganisms facilitate soil remediation in various tree species. Notably, Jm might eliminate detrimental substances from the soil via the abundant presence of Sphingomonadaceae while simultaneously harnessing the autotrophic potential of *Leptolyngbya_EcFYyyy* to sequester carbon within the soil. Consequently, the cultivation of Jm warrants consideration in instances of nutrient deficiency and ecological rehabilitation following heavy metal contamination. The primary microorganisms found in Fm facilitate the cycling and utilization of nutrients between vegetation and soil via mycorrhizal symbiosis with plants. Am, on the other hand, could enhances the carbon and nitrogen levels in the soil through the presence of nitrogen-fixing microorganisms such as *Gemmatimonas* and *Sphingomonas*. However, it is essential to note that not all tree species contribute positively to soil fertility restoration. A significant abundance of plant pathogen biomarkers enriched in Bp exhibited a

robust inverse relationship with potash levels, suggesting that their microbial activities extensively depleted soil potassium content, thereby hindering the effective accumulation of potassium in the soil. While all four tree species investigated in this study possess their respective core microorganisms with potentially beneficial functions, it is vital to prioritize the objective of this research, which aims to identify shelterbelt tree species that enhance soil fertility in the Mollisols region. Based on the comprehensive synthesis of soil nutrient content and enzyme activity findings, it is strongly recommended that Fm be extensively employed as a shelterwood species to restore soil fertility in the northeast plain.

## Conclusion

Determining the response of soil microbes to different shelterbelt tree species can help to understand plant-driven soil functioning at the micro-scale and provide information on the role of vegetation restoration in regulating soil nutrient cycling. This study analyzed the differences between soil physicochemical properties, enzyme activities, microbial communities, and the linkages among four shelterbelts through controlled experiments. We clarified that the mechanisms of soil restoration through microbial-mediated restoration differ among tree species. The results showed significant differences in soil properties among different tree species in the understory, with Fm being the most effective in ameliorating soil nutrients. The microbial community structure also changed with varying tree species, with the bacteria being more sensitive than fungi. In addition, the microbial communities recruited and colonized under the soil of each stand differed significantly due to tree species. Based on the findings above, we propose the initial selection of the Fm as the optimal tree species for establishing protection forests in northeast China. This recommendation is justified by the fact that the Fm exhibits the most comprehensive improvement in soil nutrient levels. At the same time, its core microbiota predominantly comprises probiotics that are less susceptible to environmental influences, aligning with the principles of natural ecological restoration. Furthermore, Fm demonstrates superior timber growth and possesses specific economic value, yielding both ecological and economic benefits. However, the applicability of this conclusion to long-term experiments remains to be demonstrated, and time-scale factors are critical to future research. In conclusion, this study refines the effects of tree species on microbiota and provides technical guidance for tree species selection and Mollisols restoration strategies in northeast shelterbelts.

## ACKNOWLEDGMENTS

This research was supported by the Key Project of the China National Key Research and Development Program (2021YFD150070506).

## AUTHOR AFFILIATION

[1]School of Forestry, Northeast Forestry University, Harbin, China

## AUTHOR ORCIDs

Huiyan Gu (iD) http://orcid.org/0000-0002-5718-2744

## FUNDING

| Funder | Grant(s) | Author(s) |
| --- | --- | --- |
| MOST \| National Key Research and Development Program of China (NKPs) | 2021YFD150070506 | Huiyan Gu |

## AUTHOR CONTRIBUTIONS

Jia Yang, Conceptualization, Methodology, Writing – original draft, Writing – review and editing | Dang Ding, Investigation, Methodology | Xiuru Zhang, Methodology, Resources | Huiyan Gu, Conceptualization, Funding acquisition, Project administration

## ADDITIONAL FILES

The following material is available online.

### Supplemental Material

**Tables S1 to S5 (Spectrum03683-23-s0001.docx).** Bioinformatics processing data.

### Open Peer Review

**PEER REVIEW HISTORY (review-history.pdf).** An accounting of the reviewer comments and feedback.

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
