## [Reviewer comments · Microbiology Spectrum]

Microbiology Spectrum

A Comparative Analysis of Soil Physicochemical Properties and Microbial Community Structure among Four Shelterbelt Species in the Northeast China Plain

Jia Yang, Dang Ding, Xiuru Zhang, and Huiyan Gu

Corresponding Author(s): Huiyan Gu, Northeast Forestry University

Review Timeline:

Submission Date:	October 18, 2023
Editorial Decision:	November 28, 2023
Revision Received:	January 15, 2024
Editorial Decision:	January 16, 2024
Revision Received:	January 17, 2024
Accepted:	January 23, 2024

Editor: Frédérique Reverchon

Reviewer(s): The reviewers have opted to remain anonymous.

Transaction Report:

DOI: <https://doi.org/10.1128/spectrum.03683-23>

Re: Spectrum03683-23 (Comparison of soil physicochemical and microbial of four shelterbelt species in the Northeast China Plain)

Dear Dr. Huiyan Gu:

Thank you for the privilege of reviewing your work. Below you will find my comments, instructions from the Spectrum editorial office, and the reviewer comments.

I have now received comments made by two independent reviewers on your manuscript. They both recommend modifications before your article can be accepted. As you will see below, they recommend a more detailed description of the methods, including stating the availability of the data in the Methods section, and a revision of the Discussion to ensure that your conclusions are fully supported. Also, they suggest to avoid subjective terms such as "worse" or "better" when discussing the results.

Revision Guidelines

Sincerely,
Frédérique Reverchon
Editor
Microbiology Spectrum

Reviewer #1 (Public repository details (Required)):

Genebank, the data are not available yet. The authors do not provide accession numbers.

Reviewer #1 (Comments for the Author):

The present work analyzes the microbiome, physicochemical variables, and enzymatic activities of the soil associated with four species frequently used in shelterbelts in the Northeast China Plain. The study is potentially interesting, but its applications are very local from how it is written, and its conclusions are reduced to a particular context. The biggest drawback of this article is the lack of methodological information that allows us to clearly understand which soil stratum is associated with these plant species they refer to. In general, the description of sampling and the methods for processing samples is very brief. Furthermore, during the presentation of their results and discussion, they make qualitative value judgments without giving reference criteria or specifying what they are referring to; the authors say "is worse," "is better," and "is the best," but do not indicate why, how or in which context? It is necessary to discuss the functional role of the fungal and bacterial genera found in each species to a greater extent. Finally, some of their conclusions are not based on data generated from the present work. Below, I give some specific recommendations:

Title

Please specify. Do you mean soil physicochemical properties? Microbial community structure?

Abstract

Line 15: Specifying the study's objective before describing what the authors did is necessary.

The result and interpretation of the LEfSe analysis in lines 25-27 are unclear. Why is BP not included in the fungal biomarkers?

Introduction

The term "Blackland" needs to be clarified. Does it refer to black soil in general or a specific region? It would be good if it were associated with the international pedological classification of this type of soil.

Line 63: Authors should also break down the abbreviations in the introduction, not just write the abbreviations.

Line 87-91 I understand the difficulties and limitations of this type of study. However, how the authors express it suggests that applications are at the very local level and only have these conclusions at the experimental site. Why this information should be published in an international journal if it only applies to a specific case? They must highlight the relevance of their study.

Line 100 Do you consider any hypotheses at a functional level or about the effects on amelioration?

Methods

Line 110: Where are the plants in a 3x3 m quadrant? The spatial dimension of the study needs to be explained. This section requires a more precise explanation of the study design. Was sampling random? How many replicas were there? Do you have the soil without vegetation as a reference?

Line 112-115 They do not specify how they took the soil samples. Were they taken at a certain depth or volume? Is the area under the canopy or in the rhizosphere?

Line 117-127 I suggest the authors provide more specifications on how they did the analysis.

Line 138 bioinformatics analysis instead of statistical analysis

Line 157 and 163 Replace 16S with 16S rRNA

This section needs more information on the criterion to establish OTUs; what % identity? In the case of ITS, the alignment is very complex, so a percentage is not usually used to establish OTUs as in the case of bacteria. What criteria did the authors use in this case?

Line 167-180 Specify which programs were used and provide a reference.

Results

Line 187-189: This plant seems slower-growing than the rest. I don't understand why "its growth was worse." In methods, it should be mentioned whether all these species have the same life history and similar expected development times so that these trials can be made.

Line 208-210 The criteria are unclear to give a qualitative rating of "better" or "worse" in its results.

Line 243-244 Given the many unassigned sequences at the phylum level, I suspect biases or errors in the classification method. How do the authors verify that they are not sequencing artifacts? The discussion does not mention this enormous proportion of unidentified sequences. Were these unclassified sequences used in subsequent analyses or discarded? Validation data for the RDA model are missing since the groupings described in the text are not visible in the figure.

Discussion

Line 309 "Better Physical structure" The author uses qualifiers and adjectives, but the parameters to make their assertions are not understood. This work did not measure soil structure variables such as aggregates or others.

Line 324 There is an inappropriate use of the terms. Do you mean nutrient cycling in soil? Organic matter?

Line 418-420 What are those unique characteristics? These characteristics and recommendations for using these species should be summarized.

The authors should discuss the functional role of each species' microbial communities and the advantages of using each plant

species in the shelterbelts.

Conclusions

Line 426-428 The authors mention that the mechanisms of soil restoration through microbial-mediated restoration differ among tree species; however, they do not discuss these differences in mechanisms and the applications each plant species could have in restoration.

In line 432, the authors mention that the physiological differences of each species give the differences in the physicochemical properties and that this affects the recruitment of microorganisms. However, this cannot be part of their conclusions because these variables were not part of the study or discussed in depth.

Line 436 They conclude that FM has the most "stable" microbiome but do not discuss or present data that could support this asseveration.

Reviewer #2 (Comments for the Author):

The work by Yang and colleagues analyzed four species used for shelterbelts in China, evaluating chemical and microbiological data aiming to understand the ecological importance of each species. The results suggested *Fraxinus mandschurica* as the most recommended species for that.

Although the paper is well-written and presents valuable findings, I recommend a careful review of the methods to cite all tools that were used with the proper paper being mentioned.

Minor reviews:

Line 63 - Please write the full name of the species in the first time it appears in the text, not considering the abstract.

Lines 227-245 - Please indicate the standard deviation of the mean.

Lines 623-625 - Indicate the standard deviation of the mean for tree height and diameter at breast height.

Revision

The present work analyzes the microbiome, physicochemical variables, and enzymatic activities of the soil associated with four species frequently used in shelterbelts in the Northeast China Plain. The study is potentially interesting, but its applications are very local from how it is written, and its conclusions are reduced to a particular context. The biggest drawback of this article is the lack of methodological information that allows us to clearly understand which soil stratum is associated with these plant species they refer to. In general, the description of sampling and the methods for processing samples is very brief. Furthermore, during the presentation of their results and discussion, they make qualitative value judgments without giving reference criteria or specifying what they are referring to; the authors say “is worse,” “is better,” and “is the best,” but do not indicate why, how or in which context? It is necessary to discuss the functional role of the fungal and bacterial genera found in each species to a greater extent. Finally, some of their conclusions are not based on data generated from the present work. Therefore, to be considered for publication, I recommend major revisions and resubmission. Below, I give some specific recommendations:

Title

Please specify. Do you mean soil physicochemical properties? Microbial community structure?

Abstract

Line 15 It is necessary to specify the study’s objective before describing what the authors did.

Lines 25-27 LEfSe analysis result and interpretation are unclear; why is BP not included in the fungal biomarkers?

Introduction

The term “Blackland” needs to be clarified. Does it refer to black soil in general or a specific region? It would be good if it were associated with the international pedological classification of this type of soil.

Line 63 Authors should also break down the abbreviations in the introduction, not just write the abbreviations.

Line 87-91 I understand the difficulties and limitations of this type of study. However, how the authors express it suggests that applications are at the very local level and only have these conclusions at the experimental site. Why this information should be published in an international journal if it only applies to a specific case? They must highlight the relevance of their study.

Line 100 Do you consider any hypotheses at a functional level or about the effects on amelioration?

Methods

Line 110 The plants were in a 3x3 m quadrant? The spatial dimension of the study needs to be explained. This section requires a more precise explanation of the study design. Was sampling random? How many replicas were there? Do you have the soil without vegetation as a reference?

Line 112-115 They do not specify how they took the soil samples. Were they taken at a certain depth or volume? Is the area under the canopy or in the rhizosphere?

Line 117-127 I suggest the authors provide more specifications on how they did the analysis.

Line 138 bioinformatics analysis instead of statistical analysis

Line 157 and 163 Replace 16S with 16S rRNA

This section needs more information on the criterion to establish OTUs; what % identity? In the case of ITS, the alignment is very complex, so a percentage is not usually used to establish OTUs as in the case of bacteria. What criteria did the authors use in this case?

Line 167-180 Specify which programs were used and provide a reference.

Results

Line 187-189: This plant seems slower-growing than the rest. I don't understand why "its growth was worse." In methods, it should be mentioned whether all these species have the same life history and similar expected development times to be able to make these trials.

Line 208-210 The criteria are unclear to give a qualitative rating of "better" or "worse" in its results.

Line 243-244 Given the many unassigned sequences at the phylum level, I suspect biases or errors in the classification method. How do the authors verify that they are not sequencing artifacts? The discussion does not mention this enormous proportion of unidentified sequences. Were these unclassified sequences used in subsequent analyses or discarded? Validation data for the RDA model are missing since the groupings described in the text are not visible in the figure.

Discussion

Line 309 "Better Physical structure" The author uses qualifiers and adjectives, but the parameters to make their assertions are not understood. This work did not measure soil structure variables such as aggregates or others.

Line 324 There is an inappropriate use of the terms. Do you mean nutrient cycling in soil? Organic matter?

Line 418-420 What are those unique characteristics? These characteristics and recommendations for using these species should be summarized.

The authors should discuss the functional role of each species' microbial communities and the advantages of using each plant species in the shelterbelts.

Conclusions

Line 426-428 The authors mention that the mechanisms of soil restoration through microbial-mediated restoration differ among tree species; however, they do not discuss these differences in mechanisms and the applications each plant species could have in restoration.

In line 432, the authors mention that the physiological differences of each species give the differences in the physicochemical properties and that this affects the recruitment of microorganisms. However, this cannot be part of their conclusions because these variables were not part of the study or discussed in depth.

Line 436 They conclude that FM has the most "stable" microbiome but do not discuss or present data that could support this asseveration.

Manuscript Spectrum03683-23-R1

Response letter

Dear Editor Frédérique Reverchon,

2024 Happy New Year!

We express our gratitude for being granted the opportunity to resubmit our revised manuscript titled "A Comparative Analysis of Soil Physicochemical Properties and Microbial Community Structure among Four Shelterbelt Species in the Northeast China Plain" to *Microbiology Spectrum*. We would like to acknowledge the effort you invested in handling our manuscript and the invaluable feedback and recommendations provided by the reviewers and yourself. These inputs have played a crucial role in enhancing the quality of our manuscript. Consequently, we are delighted to resubmit our article for your kind consideration.

We have diligently addressed all the comments and suggestions provided by both you and the reviewers, and we trust that our edits and the subsequent responses provided below adequately resolve all the issues and concerns raised. The manuscript has been appropriately amended, and a comprehensive point-by-point response to the reviewers' comments is presented below:

Responds to Reviewer 1:

We express our sincere gratitude for thoroughly reviewing this manuscript and offering your insightful feedback across eight distinct sections: data, title, abstract, preface, methods, results, discussion, and conclusion. We sincerely regret any lack of clarity and had questionable analytical methods in our initial submission, which may have confused the study's significance. In light of your concerns, we have diligently examined each aspect of the study and implemented substantial revisions to enhance the manuscript's coherence. The primary

objective of this paper was to improve the clarity and significance of the study by making adjustments to its content and structure. These adjustments primarily involved reanalyzing the ITS data, the removal of excessively utilized qualitative terms, and emphasizing the inherent connections between tree species and microbial communities. It is our sincere hope that these revisions will effectively elucidate the purpose and importance of this manuscript. The main revisions implemented are outlined below:

Public repository details (Required)

[Comment 1]: Genbank, the data are not available yet. The authors do not provide accession numbers.

[Our response 1]: Thanks for the heads up. The whole genome sequencing data generated in this study have been submitted to the NCBI SRA database (<https://www.ncbi.nlm.nih.gov/bioproject/>) under accession number PRJNA1047323, reviewer's accession link: <https://dataview.ncbi.nlm.nih.gov/object/PRJNA1047323>. We added this information in lines 165-167.

Title

[Comment 1]: Please specify. Do you mean soil physicochemical properties? Microbial community structure?

[Our response 1]: Thank you for your specific suggestion. We have revised the title to " A Comparative Analysis of Soil Physicochemical Properties and Microbial Community Structure among Four Shelterbelt Species in the Northeast China Plain."

Abstract

[Comment 1]: Line 15: Specifying the study's objective before describing what the authors did is necessary.

[Our response 1]: Thank you for your specific suggestion. We have added the study's aim, " Consequently, the objective is to compare the variations in soil physicochemical properties and microbial community structure within the understory of diverse shelterbelt species." in lines 16-18.

[Comment 2]: The result and interpretation of the LEfSe analysis in lines 25-27 are unclear. Why is BP not included in the fungal biomarkers?

[Our response 2]: Thank you for your question. In the LEfSe analysis, our screening conditions were at the phylum to genus level, LDA > 3. We reanalyzed the data from all ITS sequencing results and changed their LEfSe results in the abstract. We supplemented the screening conditions for the LEfSe analysis in line 29.

Introduction

[Comment 1]: The term "Blackland" needs to be clarified. Does it refer to black soil in general or a specific region? It would be good if it were associated with the international pedological classification of this type of soil.

[Our response 1]: According to the international classification standard, the object of our research is Mollisols. We have changed the expression involving soil properties and Blackland to Mollisols in the text. Blackland mainly exists in Chinese sayings that we will be more rigorous in future studies.

[Comment 2]: Line 63: Authors should also break down the abbreviations in the introduction, not just write the abbreviations.

[Our response 2]: Thank you for highlighting this problem in our manuscript. We have added the full name of the species in lines 65-66.

[Comment 3]: Line 87-91 I understand the difficulties and limitations of this type of study. However, how the authors express it suggests that applications are at the very local level and only have these conclusions at the experimental site. Why this information should be published in an international journal if it only applies to a specific case? They must highlight the relevance of their study.

[Our response 3]: I would like to apologize for any confusion caused by my lack of clarity regarding the experimental design, which may have given the impression that our experimental site was of limited size and that the results obtained were not representative. It is

important to note that our experimental sites encompass a total area of 1 hectare, with each species occupying one-quarter of the site. The planting density is clearly depicted in the experimental plots presented in Figure 1. In order to further elucidate the wide range of practical applications of this experiment in selecting on-farm shelterbelts, we have made revisions to lines 110-115 in the Methods section of the paper.

[Comment 4]: Line 100 Do you consider any hypotheses at a functional level or about the effects on amelioration?

[Our response 4]: Yes, we reintroduce hypothesis 3 that due to the different tree species, the predominant taxa manifest dissimilar functions, thereby resulting in disparities in their impact on soil enhancement. This change occurs in lines 100-101 of the revised manuscript.

Methods

[Comment 1 & 2]: Line 110: Where are the plants in a 3x3 m quadrant? The spatial dimension of the study needs to be explained. This section requires a more precise explanation of the study design. Was sampling random? How many replicas were there? Do you have the soil without vegetation as a reference?

Line 112-115 They do not specify how they took the soil samples. Were they taken at a certain depth or volume? Is the area under the canopy or in the rhizosphere?

[Our responses 1 & 2]: We sincerely apologize for the insufficiency of our description regarding the experimental methodology. It is essential to clarify that the plants were not planted in 3m*3m quadrants but with a 3m spacing between each other. Furthermore, during 2022, the crop intercropped under the four stands consisting of soybeans. Nearby, within a 2km radius of the sample plots, the pure cropland crops consisted of cabbage and potatoes. Consequently, we conducted the same physicochemical experiments and microbial sequencing on the soil of the cabbage plot. However, due to crop disparities, we refrained from employing soil without plants as a control in this experiment.

[Our acts 1 & 2]: In April 2019, four sample plots measuring 25m*100m were established on a one-hectare cropland area. Each plot was planted with identical plants, spaced 3m apart (Jm, Fm, Am, and Bp). Subsequently, in October 2022, each sample plot was randomly divided

into three smaller sample squares measuring 5m*5m. A five-point sampling method was employed in each sample plot to collect the top layer of bulk soil (0-10cm) after removing surface litter and debris. A subset of these soil samples was then sieved and promptly frozen for DNA extraction, while the remaining soil samples were air-dried and analyzed for chemical properties and enzymatic activity. In addition, tree height and diameter at breast height (DBH) were measured as indicators of plant growth.

[Comment 3]: Line 117-127 I suggest the authors provide more specifications on how they did the analysis.

[Our response 3]: We have added the specific experimental methodology in lines 120-130.

[Comment 4]: Line 138 bioinformatics analysis instead of statistical analysis

[Our response 4]: We have replaced statistical analysis in line 149 with bioinformatics analysis.

[Comment 5]: Line 157 and 163 Replace 16S with 16S rRNA

[Our response 5]: We have replaced 16S in lines 172 & 178 with 16S rRNA.

[Comment 6]: This section needs more information on the criterion to establish OTUs; what % identity? In the case of ITS, the alignment is very complex, so a percentage is not usually used to establish OTUs as in the case of bacteria. What criteria did the authors use in this case?

[Our response 6]: Thank you for your inquiry regarding data processing. The canonical clustering threshold was 97% identity, added in line 168. Initially, in the preliminary version, we employed the identical clustering technique (UPARSE) for fungi and bacteria in the data processing phase. However, in response to your valuable feedback, we reanalyzed the ITS gene amplicon fragment datasets utilizing a denoising approach (DADA2). Subsequently, all outcomes derived from this analytical method were visually reconstructed, and the corresponding results and discussion sections were thoroughly revised.

[Comment 7]: Line 167-180 Specify which programs were used and provide a reference.

[Our response 7]: Our data is analyzed online using a cloud platform, not an R package. We have provided relevant references in line 170, and we hope this addresses the questions you have raised.

s

Results

[Comment 1]: Line 187-189: This plant seems slower-growing than the rest. I don't understand why "its growth was worse." In methods, it should be mentioned whether all these species have the same life history and similar expected development times so that these trials can be made.

[Our response 1]: We appreciate your observation regarding the ambiguity. We intend to convey that the wood volume differs across stands with identical site conditions and that this volume is determined by the diameter at breast height and tree height. In lines 201-202, we made an amendment to elaborate on this result. This research aims to establish a scientific framework for selecting tree species when converting plantation protection forests to plantation economic forests. Additionally, we will exercise caution in employing exaggerated language to describe the findings.

[Comment 2]: Line 208-210 The criteria are unclear to give a qualitative rating of "better" or "worse" in its results.

[Our response 2]: I regret the excessive use of qualitative terms in my previous statement. The data presented in Table 1 and Figure 2 shows that the highest values were recorded for soil enzyme activity and chemical nutrients, leading to the conclusion that Fm outperformed the other species in this regard. This amendment has been made in lines 223-225.

[Comment 3]: Line 243-244 Given the many unassigned sequences at the phylum level, I suspect biases or errors in the classification method. How do the authors verify that they are not sequencing artifacts? The discussion does not mention this enormous proportion of unidentified sequences. Were these unclassified sequences used in subsequent analyses or discarded? Validation data for the RDA model are missing since the groupings described in the text are not visible in the figure.

[Our response 3]: We express our gratitude for your meticulous examination of the data. We reanalyzed the ITS gene amplicon fragment datasets utilizing a denoising approach (DADA2). Subsequently, all outcomes derived from this analytical method were visually reconstructed, and the corresponding results and discussion sections were thoroughly revised.

Discussion

[Comment 1]: Line 309 "Better Physical structure" The author uses qualifiers and adjectives, but the parameters to make their assertions are not understood. This work did not measure soil structure variables such as aggregates or others.

[Our response 1]: We sincerely appreciate your valuable input in identifying the logical inconsistency during the discussion. To address this concern, we have provided a more precise description instead of the physical structure and bolstered the reliability of the logical argument by incorporating relevant references. The revised version can be found in lines 328-335.

[Comment 2]: Line 324 There is an inappropriate use of the terms. Do you mean nutrient cycling in soil? Organic matter?

[Our response 2]: The term "cycling" in this context specifically pertains to the carbon, nitrogen, phosphorus, and sulfur cycles, which hold ecological significance. The additional information has been incorporated into lines 348-349.

[Comment 3]: Line 418-420 What are those unique characteristics? These characteristics and recommendations for using these species should be summarized.

[Comment 4]: The authors should discuss the functional role of each species' microbial communities and the advantages of using each plant species in the shelterbelts.

[Our response 3&4]: We express gratitude for the guidance. The distinctive attributes mentioned possess the capability to serve as functions of their respective core microorganisms. We thoroughly examine the functionality of the microbial community associated with each species and consolidate the benefits of employing each species as the shelterbelt. The inclusion above can be found within the revised manuscript, specifically on lines 475-497.

Conclusions

[Comment 1]: Line 426-428 The authors mention that the mechanisms of soil restoration through microbial-mediated restoration differ among tree species; however, they do not discuss these mechanisms and the applications each plant species could have in restoration.

[Our response 1]: We regret the lack of clarity in our previous mention of mechanisms and acknowledge the need for further elucidation. In order to address this, we have provided a comprehensive summary of the mechanisms involved in microbial soil remediation, specifically those mediated by various forest stands, in the discussion section (lines 475-497). We hope this revision will enhance the reader's comprehension of our experimental findings and their associated implications.

[Comment 2]: In line 432, the authors mention that the physiological differences of each species give the differences in the physicochemical properties and that this affects the recruitment of microorganisms. However, this cannot be part of their conclusions because these variables were not part of the study or discussed in depth.

[Our response 2]: We express our gratitude for the valuable suggestions for the conclusion section's content. In response, we have eliminated any conclusions that are not directly derived from the results of this experiment and have made revisions to this section as indicated in lines 508-509.

[Comment 3]: Line 436 They conclude that FM has the most "stable" microbiome but do not discuss or present data that could support this asseveration.

[Our response 3]: We appreciate your inquiry regarding words with ambiguous meanings. In response, we have substituted the term "stable" with a more precise sentence elucidating the characteristics of Fm soil microbes. Previous research has demonstrated a negative correlation between soil quality and the composition of its microbial community (Yang et al., 2023). Specifically, a lower number of biomarkers screened indicates a reduced quantity, while "beneficial" refers to taxa scientifically proven to facilitate soil restoration and nutrient cycling. In the context of this study, Fm aligns with this pattern by exhibiting the fewest

biomarkers, predominantly consisting of probiotics with established functional properties. This modification has been made in lines 511-516.

Responds to Reviewer 2:

We appreciate your careful reading of our paper and your positive comments. We have made the following corrections to address the issues you raised.

[Comment 1]: Line 63 - Please write the full name of the species the first time it appears in the text, not considering the abstract.

[Our response 1]: We have added the full name of the species in lines 65-66.

[Comment 2]: Lines 227-245 - Please indicate the standard deviation of the mean.

[Our response 2]: Due to the limited length of the article, we wanted to highlight more valid information in the main text. Therefore, the standard deviation of the mean was listed in Table S2&S3, and we hope that this revision satisfies you.

[Comment 3]: Lines 623-625 - Indicate the standard deviation of the mean for tree height and diameter at breast height.

[Our response 3]: We have added the mean for tree height and diameter at breast height in Table 1.

YANG J, HE J, JIA L, et al. 2023. Integrating metagenomics and metabolomics to study microbiota's response in black soil degradation. *Science of The Total Environment* [J], 899: 165486.

Re: Spectrum03683-23R1 (A Comparative Analysis of Soil Physicochemical Properties and Microbial Community Structure among Four Shelterbelt Species in the Northeast China Plain)

Dear Dr. Huiyan Gu:

Thank you for the privilege of reviewing your work. Below you will find my comments, instructions from the Spectrum editorial office, and the reviewer comments.

I have evaluated the revised manuscript and, while I am satisfied with the authors' response to the concerns raised by the reviewers, I believe there are several issues that still need to be attended.

L21: replace "and physicochemical" experiment" by "and soil physicochemical data"

L30: do you mean that the relative abundance of these taxa was altered?

L31-35: these two sentences are contradictory. The first indicate that soil properties drive soil microbial communities, the second indicates that microbial communities influence Mollisols properties. I suggest to avoid mentioning causal effects in the manuscript as RDA or correlations do not indicate a causal relationship.

L92-93: "Therefore, it is necessary to utilize microbial diversity to study the response of soil microbes to different tree species". I disagree with this statement. Soil microbial diversity as a metrics may not be affected by management, however microbial community composition may be altered. So diversity may not be a useful indicator of how "soil microbes" respond to different tree species.

L94: please remove "a class of", as "class" is also a taxonomical level.

L97: I disagree, not all studies consider large spatial scale nor integrate many environmental factors, and most of them control for this variation. Saying that their results are inaccurate is a strong statement.

L157: instead of "bioinformatic analyses", please use "library construction". You are not analysing anything yet.

L333: instead of "mechanisms", please use "relationships". Mechanisms may only be inferred as no causal effect can be deduced from correlations.

L391-392: not always, many studies have also attributed the differences in taxa relative abundance to the plant physiological stage (health, age) or genotype. I think this statement excludes many other investigations and is not entirely true. Stating that different tree species harboured different bacterial communities is not new and examples should be provided.

L394: "This result helps to provide research ideas for the microbial mechanisms of soil remediation by Jm and Bp." How so? Please elaborate or remove this statement.

L402: some members of Mortierella may also be plant pathogens, not all species are plant growth promoters.

L444: I believe Fraxinus forms arbuscular mycorrhiza and not ectomycorrhiza.

L461, L465 and onward: again, I do not think you can describe causal effects based on RDA or correlations. Please mitigate your statements. For example, L487: "Furthermore, Fm and Am employ Metarhizium and Schizothecium, respectively, to protect against the proliferation of soil-borne pathogens". It is a hypothesis that has not be tested, you should rephrase this statement. These genera may be actively recruited by the plant or may not, regardless of their positive (but untested here) role in pathogen inhibition.

Revision Guidelines

- Upload point-by-point responses to the issues raised by the reviewers in a file named "Response to Reviewers," NOT IN

YOUR COVER LETTER

- Upload a compare copy of the manuscript (without figures) as a "Marked-Up Manuscript" file
- Upload a clean .DOC/.DOCX version of the revised manuscript and remove the previous version
- Each figure must be uploaded as a separate, editable, high-resolution file (TIFF or EPS preferred), and any multipanel figures must be assembled into one file
- Any supplemental material intended for posting by ASM should be uploaded separate from the main manuscript; you can combine all supplemental material into one file (preferred) or split it into a maximum of 10 files, with all associated legends included

Sincerely,
Frédérique Reverchon
Editor
Microbiology Spectrum

Response letter

Dear Editor Frédérique Reverchon,

We express our gratitude for your approval of our initial revision. Additionally, we appreciate your thorough review and subsequent identification of specific concerns regarding the details and expressions within the paper. This constructive feedback has greatly contributed to enhancing the comprehensiveness and objectivity of our article. We have duly addressed these concerns by appropriately amending the manuscript. A comprehensive point-by-point response to your comments is provided below:

[Comment 1]: L21: replace "and physicochemical" experiment" by "and soil physicochemical data"

[Our response 1]: We have replaced "physicochemical" by "soil physicochemical data" in L20.

[Comment 2]: L30: do you mean that the relative abundance of these taxa was altered?

[Our response 2]: Yes. We added relative abundance in L29.

[Comment 3]: L31-35: these two sentences are contradictory. The first indicate that soil properties drive soil microbial communities, the second indicates that microbial communities influence Mollisols properties. I suggest to avoid mentioning causal effects in the manuscript as RDA or correlations do not indicate a causal relationship.

[Our response 3]: Thank you for your advices. We have replaced "key drivers of" by "important factors influencing" in L32. Furthermore, we deleted the second sentence in Reversion 2.

[Comment 4]: L92-93: "Therefore, it is necessary to utilize microbial diversity to study the response of soil microbes to different tree species". I disagree with this statement. Soil

microbial diversity as a metrics may not be affected by management, however microbial community composition may be altered. So diversity may not be a useful indicator of how "soil microbes" respond to different tree species.

[Our response 4]: We rephrased this sentence by "studying the response of microbial communities to different tree species helps us to understand soil microecology" in L85-86.

[Comment 5]: L94: please remove "a class of", as "class" is also a taxonomical level.

[Our response 5]: We have deleted "a class of" in Reversion 2.

[Comment 6]: L97: I disagree, not all studies consider large spatial scale nor integrate many environmental factors, and most of them control for this variation. Saying that their results are inaccurate is a strong statement.

[Our response 6]: I apologize for the arbitrary presentation and have deleted my subjective evaluation of other papers throughout this text in Reversion 2.

[Comment 7]: L157: instead of "bioinformatic analyses", please use "library construction". You are not analysing anything yet.

[Our responses 7]: We have replaced "bioinformatic analyses" by "library construction" in L148.

[Comment 8]: L333: instead of "mechanisms", please use "relationships". Mechanisms may only be inferred as no causal effect can be deduced from correlations.

[Our response 8]: We have replaced "mechanisms" by "relationships" in L319.

[Comment 9]: L391-392: not always, many studies have also attributed the differences in taxa relative abundance to the plant physiological stage (health, age) or genotype. I think this statement excludes many other investigations and is not entirely true. Stating that different tree species harboured different bacterial communities is not new and examples should be provided.

[Our response 9]: We have deleted our subjective evaluation in Reversion 2.

[Comment 10]: L394: "This result helps to provide research ideas for the microbial mechanisms of soil remediation by Jm and Bp." How so? Please elaborate or remove this statement.

[Our response 10]: We have deleted this statement in Reversion 2.

[Comment 11]: L402: some members of *Mortierella* may also be plant pathogens, not all species are plant growth promoters.

[Our response 11]: We have replaced "also" by "some species" in L382.

[Comment 12]: L444: I believe *Fraxinus* forms arbuscular mycorrhiza and not ectomycorrhiza.

[Our response 12]: Thank you for point out this key mistake. We have modified "ectomycorrhizal" by "arbuscular mycorrhizal" in L424.

[Comment 13]: L461, L465 and onward: again, I do not think you can describe causal effects based on RDA or correlations. Please mitigate your statements. For example, L487: "Furthermore, *Fm* and *Am* employ *Metarhizium* and *Schizothecium*, respectively, to protect against the proliferation of soil-borne pathogens". It is a hypothesis that has not be tested, you should rephrase this statement. These genera may be actively recruited by the plant or may not, regardless of their positive (but untested here) role in pathogen inhibition.

[Our response 13]: 1) We have modified "effect" by "conditional effect" in L440; 2) We have modified "roles of" by "relationship with" in L444; 3) We have deleted this statement in Reversion 2.

Re: Spectrum03683-23R2 (A Comparative Analysis of Soil Physicochemical Properties and Microbial Community Structure among Four Shelterbelt Species in the Northeast China Plain)

Dear Dr. Huiyan Gu:

I am satisfied with the revisions and can now proceed to accept the manuscript.

Please make sure to correct in the final version of your manuscript, line 449 (marked-up manuscript): Clitopilus and Amanita are indeed ectomycorrhizal fungi. However, there are enlisted, line 446, as potential symbionts of Fraxinus, but Fraxinus forms a symbiosis with arbuscular, not ectomycorrhizal fungi.

Your manuscript has been accepted, and I am forwarding it to the ASM production staff for publication. Your paper will first be checked to make sure all elements meet the technical requirements. ASM staff will contact you if anything needs to be revised before copyediting and production can begin. Otherwise, you will be notified when your proofs are ready to be viewed.

Sincerely,
Frédérique Reverchon
Editor
Microbiology Spectrum